# The Impact of Yeast Encapsulation in Wort Fermentation and Beer Flavor Profile

**DOI:** 10.3390/polym15071742

**Published:** 2023-03-31

**Authors:** Angie D. Bolanos-Barbosa, Cristian F. Rodríguez, Olga L. Acuña, Juan C. Cruz, Luis H. Reyes

**Affiliations:** 1Product and Process Design Group (GDPP), Department of Chemical and Food Engineering, Universidad de Los Andes, Bogotá 111711, Colombia; 2Department of Biomedical Engineering, Universidad de Los Andes, Bogotá 111711, Colombia

**Keywords:** alginate encapsulation, beer brewing, alcoholic fermentation, flavor modification, sensory profile

## Abstract

The food and beverage industry is constantly evolving, and consumers are increasingly searching for premium products that not only offer health benefits but a pleasant taste. A viable strategy to accomplish this is through the altering of sensory profiles through encapsulation of compounds with unique flavors. We used this approach here to examine how brewing in the presence of yeast cells encapsulated in alginate affected the sensory profile of beer wort. Initial tests were conducted for various combinations of sodium alginate and calcium chloride concentrations. Mechanical properties (i.e., breaking force and elasticity) and stability of the encapsulates were then considered to select the most reliable encapsulating formulation to conduct the corresponding alcoholic fermentations. Yeast cells were then encapsulated using 3% (*w*/*v*) alginate and 0.1 M calcium chloride as a reticulating agent. Fourteen-day fermentations with this encapsulating formulation involved a Pilsen malt-based wort and four *S. cerevisiae* strains, three commercially available and one locally isolated. The obtained beer was aged in an amber glass container for two weeks at 4 °C. The color, turbidity, taste, and flavor profile were measured and compared to similar commercially available products. Cell growth was monitored concurrently with fermentation, and the concentrations of ethanol, sugars, and organic acids in the samples were determined via high-performance liquid chromatography (HPLC). It was observed that encapsulation caused significant differences in the sensory profile between strains, as evidenced by marked changes in the astringency, geraniol, and capric acid aroma production. Three repeated batch experiments under the same conditions revealed that cell viability and mechanical properties decreased substantially, which might limit the reusability of encapsulates. In terms of ethanol production and substrate consumption, it was also observed that encapsulation improved the performance of the locally isolated strain.

## 1. Introduction

The increasing demand for beer has led several breweries to develop new styles and profiles to enable improved sensorial experiences for customers. Due to its popularity, beer has reached an annual production that exceeds 1400 million hectoliters globally [1]. It is commercially available in both industrial and craft versions. The former category is responsible for hundreds of thousands to millions of hectoliters per year, while the latter is responsible for less than 200,000 hectoliters per year. However, these volumes may vary depending on how each country categorizes the products. For example, the Brewers Association categorizes breweries into groups based on their production capacity and product distribution strategy [2].

Four ingredients are required to produce beer: malt, yeast, hops, and water. The addition of adjuncts, such as sugars or syrups, as well as novel elements such as alginate, carrageenan, iso–acids, fruits, spices, benzoic acid, and vegetable gums, among many others, can be exploited to achieve a variety of flavors and beer profiles [3,4].

Malting is the initial step in beer production, in which partially germinated grain is dried and toasted. The malt is then used in a process called mashing to produce wort, a broth rich in simple sugars. In certain instances, additional sources of complex sugars, such as rice and corn, may be used to enrich the wort. During mashing, some compounds and proteins that enrich the culture medium are solubilized, and complex sugars are enzymatically hydrolyzed to be used as a substrate by yeast [5]. The most commonly used yeast to produce alcoholic beverages is *Saccharomyces cerevisiae*. Various strains of this yeast have been employed in the brewing industry to produce different flavors for the various beer styles [6].

Depending on the style of the beer produced, various types of yeast can be employed. For instance, *S. cerevisiae* is used in ale–style beers, whereas *S. pastorianus* is used in the lager–style ones. Both types of yeast perform differently based on conditions such as temperature and pH. The first grows at 20 °C to 30 °C and at pH levels between 4.5 and 6.5. In comparison, the second grows at temperatures between 8 °C and 15 °C. *S. cerevisiae* is a unicellular fungus that exhibits characteristics similar to other higher eukaryote cells. It shows an ellipsoid form with a long diameter between 5 and 10 μm and a short diameter of approximately 5 μm [7].

The fermentation process also requires oxygen, typically supplied by wort aeration at the beginning of the process [6], and multiple additional nutrients, including cofactors such as minerals, vitamins, inorganic ions, ammonia salts, and amino acids, among others [8]. Since secondary metabolites are produced during fermentation, not all substrate is converted into ethanol and some of these molecules may affect the flavor, aroma, viscosity, and even the texture of beers. Graenmer et al. detail the specific metabolites yeast produces during fermentation [6]. In addition to ethanol and carbon dioxide, yeast can produce aldehydes, ketones, esters, fatty and organic acids, a few sulfur compounds, and higher–order alcohols [9]. The Maillard reactions produce aldehydes in the wort, which contribute to undesirable flavors. When amino acids are metabolized, pentanedione and diacetyl are produced, which exhibit a caramel aroma [6]. Along with transamination, decarboxylation, and reduction of amino acids, pyruvate can be used to synthesize higher–order alcohols. Esters such as ethyl acetate, isoamyl acetate, and ethyl caproate can be produced through esterification reactions between alcohols (including ethanol) and different fatty acids. Typically, these compounds are associated with fruity and floral aromas [9,10].

Beer yeasts vary in performance, ability to metabolize maltose (the primary sugar found in malt), aroma production, and resistance to alcoholic environments [7]. Beer’s accessibility has led to a variety of product innovation strategies, such as the genetic modification of yeast to alter the organoleptic properties or produce a specific aroma, the addition of excipients, and the implementation of technologies to improve alcohol production, such as yeast encapsulation in calcium alginate [7,11]. This encapsulation strategy may not only impact the fermentation performance but also the production of volatile compounds, such as terpenes, phenols, and organic acids, which can result in appealing sensory profiles [12].

Alginate is a polysaccharide present in brown seaweed, such as *Macrocystis pyrifera*, *Laminaria hyperborea*, and *Ascophyllum nodosum*. Alginate is formed by alternating blocks of two monomeric units, the β–D mannuronic acid (M) and the α–L guluronic acid (G), which can hold different arrangements (i.e., MG, MM, or GG). The main physical and mechanical properties (e.g., compression resistance and rigidity) of alginate strongly depend on the arrangement of the blocks conforming the polymer chain. These properties increase in the following order: MG < MM < GG [13,14].

When using sodium alginate and calcium chloride, an ion exchange reaction between Ca^2+^ and Na^+^ happens, leading to the formation of NaCl salt and the linking of polymer chains through coordination sites formed when two GG chains are aligned, allowing the polyvalent calcium cations to interact with alginate. As a result, conformational changes in the biopolymer may occur, resulting in a structure that follows an “egg carton” model due to the chain’s positioning with respect to the cations [15]. These reactions may be completed at room conditions without other ions that can cause interference. In the case of calcium ions, there is a cooperative binding that results in a firmer structure than with other ions. The structures of guluronic acid provide a distance between hydroxyl and carboxyl groups, which confers a high degree of coordination [14].

Alginate is one of the most widely used polymers for encapsulation in the food industry because it is considered safe, inexpensive, and forms nanocontainers and microcapsules in various ways [16]. For instance, it has been used with chitosan to create nanoparticles as a vehicle for delivery of hydrophobic drugs, in conjunction with electro–spray–assisted microencapsulation techniques to slow down the release of substances such as caffeine and to encapsulate probiotic bacteria such as *Lactobacillus*, *Bifidobacterium*, and many others [17,18,19,20,21]. Ching et al. present additional encapsulating techniques, including electrostatic potential, vibrating nozzles, spray nozzles, and microfluidics, that can produce microcapsules with diameters between 0.2 mm and 1.0 mm or nanocapsules with diameters below 0.2 mm [22].

Macroscopic–size capsules (>1 mm) are suitable for immobilizing yeasts. There are numerous methods for preparing macroscopic capsules. The extrusion technique is one of the simplest, utilizing needles of varying diameters to control the size of the macroscopic spheres [22]. To encapsulate yeast, cells are mixed with an alginate solution and then dropped into the cross–linking agent. The formation of droplets is accomplished by using needles with 21 G or 18 G calibers. The diameters of the manufactured capsules range from 2 mm to 5 mm. Size and mechanical properties also depend on alginate and cross–linking agent concentrations [23,24]

Cell immobilization shows a series of advantages compared to free cells, such as the enhancement of cell stability and performance, operational flexibility, the possibility for continuous production, ease of cell recovery, and reusability [12,24,25,26]. Some techniques help to preserve cell viability, as in the case of agarose encapsulation [27]. All of this is possible because of the capsules’ enhanced resistance to adverse conditions [25,28,29]. However, in the presence of chelating agents and sodium or magnesium ions, alginate hydrogels may exhibit decreased mechanical stability and capsule liquefaction [22]. Cell leakage from the polymeric matrix through the pores, cell ingrowth within the capsules, and gas production may also have a negative effect on the capsules’ mechanical properties [28]. Among the alternatives considered to mitigate these disadvantages, using more stable cations such as Ba^2+^, Pb^2+^, or Cu^2+^ stands out; however, their toxicity to cells might be somewhat worrisome. Other attractive alternatives include co–polymerization, silica–coating, photo cross–linking, and covalent cross–linking, among others [28].

The objective of this study is to determine a suitable calcium alginate formulation for developing encapsulates containing *S. cerevisiae* yeasts for use in alcoholic fermentation processes to investigate possible alterations in the sensory profiles of finished products. The selection of the formulation proceeded by considering the mechanical, thermal, and morphological properties of the synthesized capsules. The sensory profiles were determined for Pilsen malt–based worts fermented with four different yeast strains either encapsulated or free.

## 2. Materials and Methods

The methodology is organized in two sub–sections. The first one covers the formulation and production of yeast encapsulates and the evaluation of their physical properties using multiple techniques. The second one describes all the fermentative processes involving encapsulated and free yeast. Minitab 17 (Minitab Inc., State College, PA, USA) was utilized for the statistical analysis.

### 2.1. Formulation and Characterization of Sodium Alginate Capsules

#### 2.1.1. Microorganisms and Culture Media

The *S. cerevisiae* strains selected for this study are listed in Table 1.

The first three strains are commercial, lyophilized strains that are available in 11.5 g sachets. CW–12, on the other hand, is a native Colombian strain obtained from a manufacturer in the Colombian Department of Boyacá that processes unrefined whole cane sugar. During beer production, the strain was isolated, identified, and characterized in terms of the production of flavor, aroma, alcohol, and organic acids [30].

The procedure for the commercial yeast’s pre-inoculum started with the hydration of 1 g of yeast in 40 mL of YPD medium (yeast extract 1.5% (*w*/*v*) (Sigma-Aldrich, St. Louis, MO, USA), peptone 2% (*w*/*v*) (Millipore, Milwaukee, WI, USA), and dextrose 2% (*w*/*v*) (Sigma-Aldrich, St. Louis, MO, USA) for 15 min. Once hydrated, 300 mL of YPD growing media was inoculated to an initial OD_600_ value of 0.2 for all strains (Appendix A). Subsequently, inoculated media were incubated at 30 °C and 200 rpm for 24 h. The CW–12 procedure began with removing 10 μL from the cryovial and cultivating it in a Petri dish with YPD agar at 30 °C. After transferring one colony–forming unit (CFU) from the Petri dish to a Falcon tube containing 20 mL of YPD liquid medium, the tube was incubated overnight at 30 °C and 200 rpm. This incubated liquid YPD medium served as the pre–inoculum. The procedure continued by inoculating 300 mL of YPD medium to an initial OD_600_ value of 0.2. Then such media were incubated for 24 h at 30 °C and 200 rpm (Appendix A).

After 24 h, the growing media were transferred to sterile 50 mL Falcon tubes and left to stand for 3 h at room temperature (Appendix A). Subsequently, they were centrifuged at 4 °C and 4000× *g* for 20 min in a refrigerated centrifuge (Thermo Scientific, Waltham, MA, USA). The supernatant was discarded, and the biomass was washed with Type I autoclaved water. The obtained suspension was allowed to stand at room temperature for an additional three hours before being centrifuged under the same conditions. Finally, biomass was weighed and stored for encapsulation.

#### 2.1.2. Yeast Encapsulation

The description of the process for elaborating the yeast encapsulates is presented below. As shown in Figure 1, an experimental design was established in which six formulations were chosen to vary the sodium alginate and calcium chloride concentrations. The alginate used in this study has a low M/G ratio of approximately 0.4 and a high viscosity.

According to the experimental design, the biomass obtained in the previous step was combined with sodium alginate to produce a mixture containing 20 wt. % yeast. The mixture was then transferred to a 60 mL syringe with a needle of 21 G. The assembly was completed using a syringe pump, as depicted in Appendix A. A stock solution of 0.2 M calcium chloride was prepared, followed by multiple dilutions to achieve the desired concentrations of 0.1 M and 0.05 M. Then 200 mL of the diluted solution was added to a 500 mL beaker placed on a 250 rpm magnetic stirrer. The syringe was positioned 15 cm above the surface of the chloride solution, so that the produced droplets collided with the mixing vortex as they fell into the solution.

The encapsulation procedure was completed using pulse cycles, in which the syringe dripped at 2 min intervals. The program was then paused for five minutes while the capsules were maintained under continuous stirring. They were then screened and stored in type I distilled water. The preceding procedure was repeated until the syringe’s contents were depleted. The calcium chloride solution in the beaker was replaced every five rounds to obtain uniform capsules; each round’s capsules were collected and stirred for five more minutes to ensure homogeneity. All capsules were then rinsed with autoclaved type I water and stored immersed in type I water in Schott bottles at 4 °C until further use. The procedure mentioned above was repeated for each formulation presented in the experimental design.

#### 2.1.3. Encapsulate Characterization

Size, texture, swelling percentage, spectroscopy, thermal stability, diffusivity, and morphology analyses were performed on the obtained encapsulates. Similarly, these tests were conducted on encapsulates recovered after 1, 2, and 3 repeated batch fermentations to evaluate processing–induced changes. The methods for each of the analyses are described in detail below.

*Swelling Percentage*: The swelling percentage was calculated using Equation (1), which relates the weight difference of the capsules (*W*_0_: initial weight and *W_t_:* weight at time *t*) after being immersed in the wort for a period of time. [31].
(1)$$Swelling\%=\frac{W_{t}-W_{0}}{W_{0}}\times 100\;$$

*Spectroscopic Analysis*: The main functional groups and chemical bonds present in the capsules were determined via Fourier transform infrared spectroscopy (FT–IR) on a Nicolet ^TM^ iS50 FT–IR Spectrometer (Thermo Scientific^TM^, San Francisco Bay Area, CA, USA). The spectra were collected for wavenumbers ranging from 500 to 4000 cm^−1^ with a spectral resolution of 8 and 32 scans per sample. The collection time was 27.74 s, the gain value was 1, and the opening value corresponded to 100. The detector was the DTGS ATR. Due to the use of water as the blank, no sample preparation, such as drying or lyophilization, was necessary.

*Mechanical resistance*: The encapsulates were subjected to a compression test utilizing the Gel Capsule Penetration Method on a TA HDPlusC texture analyzer (Stable Micro Systems, Godalming, UK) at a speed of 0.5 mm/s. The experiment was conducted with a P2 probe, a 5 kg cell, and a 5 g activation force.

*Thermal Stability*: Thermogravimetric analysis (TGA) and differential scanning calorimetry analysis (DSC) were performed on a Q600 Simultaneous TGA/DSC (TA Instruments, New Castle, DE, USA) to determine the thermal stability of the capsule and the amount of heat absorbed or released by the polymer. The temperature range was between 50 and 600 °C and the heating rate was 10 °C per minute. The experiment was conducted at a 100 mL/min flow rate in a non–oxidizing atmosphere controlled by ultra–high–purity nitrogen (UHP) [32].

*Structure and morphology of the encapsulates*: Because there is a pore size gradient from the center to the edge of the encapsulates, it was necessary to observe the encapsulates’ center, internal border, and external surface by slicing them in half. Before observation by scanning electron microscopy (SEM), the samples were cryopreserved with liquid nitrogen (provided by the Cryogenics Department of the University of the Andes). The morphology of the gels was observed with 30×, 500×, and 1000× magnifications at 10 kV and 20 kV acceleration voltages. At a temperature of −15 °C, measurements were conducted at the Microscopy Center of the University of the Andes using a JEOL SEM instrument (model JSM 6490–LV, Dearborn Road, Peabody, MA, USA) [6]. The images were analyzed using the ImageJ^®^ software to determine the pore size distribution and porosity [31].

*Cell viability analysis*: Cell viability was determined with a LIVE/DEAD Yeast Viability Kit (Invitrogen™, Waltham, MA, USA). The kit contains two fluorescent markers, FUN1 and White M2R Calcofluor^®^, which are specific to stain live (red) and total (blue) yeast cells, respectively [33]. FUN1 enables the identification of living yeasts because it requires the membrane integrity and metabolic function of the cells to alter the staining from yellow/green to reddish/orange, effectively. Calcofluor White M2R stains chitin in cell wall membranes blue even when the membrane’s integrity has been compromised. The encapsulates were submerged in the marker solutions for thirty minutes before observation with an Olympus FV1000 (Olympus, Tokyo, Japan) (40×, 0.6 NA) confocal laser scanning microscope. Using the Fiji–ImageJ^®^ software, the percentage of living cells was then determined utilizing the acquired images.

### 2.2. Fermentations

The Pilsen base malt wort was prepared, followed by the assembly of the fermenting recipients, which were then filled and inoculated with each yeast strain in its free or encapsulated form. The samples obtained after fermentation and maturation were sensory, chromatographically, and physically analyzed in terms of color and turbidity.

#### 2.2.1. Wort Preparation

Pilsen base malt (Bestmalz, Heidelberg, GER) was utilized in the wort’s production. According to the manufacturer, this malt can produce wort with a color between 5.8 and 6.1 [34]. Considering the activation temperatures of the various enzymes in the malt [35,36,37], the mashing was completed in a PicoBrew (model C, Kirkland, WA, USA) since it allowed for temperature control and recirculation of water. Figure 2 depicts the temperature profile established for malt mashing.

At the end of the process, the wort was transferred to amber Schott bottles and autoclaved. The wort was then cooled to room temperature and filtered under vacuum with an 8–12 μm filter (BOECO, Germany, grade 389, ref. 3.102.125 of 84 g/m^2^) to remove protein and tannin precipitates generated after sterilization. No hops were added to the mixture.

#### 2.2.2. Fermenter Assembly

The fermenter consisted of a 750 mL glass container with a hermetically sealed metallic cover, equipped with an airlock valve (to vent carbon dioxide) and a needle for sampling, as shown in Appendix A. All the fermenters were filled with 600 mL of filtered wort. Then 50 g of yeast per hectoliter was used for the inoculation (according to the instructions for commercial yeast), equivalent to an initial OD_600_ value of 0.45. To achieve similar conditions of inoculation between encapsulated and free yeast, the OD_600_ was determined for a yeast/alginate mixture (20% *w*/*v* of yeast), which allowed us to estimate that 260 capsules were needed to inoculate 600 mL of wort. Once inoculated, each fermenter was hermetically sealed. A seal on the top of the needle was removed every time sampling was performed. Each fermenter was wrapped in aluminum foil to prevent light–induced deterioration of medium components. The fermentation process at room temperature lasted fourteen days.

After the fermentation ended, the fermented wort was transferred to amber Schott bottles and refrigerated for two weeks for the maturation stage. The encapsulates were recovered from the fermenters, rinsed with distilled type I water, and stored at 4 °C in Schott bottles until further use. As soon as the maturation period concluded, tests for color, turbidity, and electronic tongue and nose were conducted swiftly to prevent undesirable oxidation reactions that might alter the sensory profile.

#### 2.2.3. Fermentation Analysis

*Yeast growth*: Yeast growth was monitored for 170 h, equivalent to the first 7 days of fermentation. Samples of 1 mL were taken from time to time to measure the OD_600_ parameter in a multi–cell UV–VIS spectrometer (Mettler Toledo, Columbus, OH, USA).

*Color and turbidity*: The methodology established in the “analytical methods for the brewing industry” manual was followed to measure color and turbidity. A spectrophotometric method was used to read the sample at 430 nm (*A*_430nm_) to calculate the color in EBC units following Equation (2) [38].
SRM = 12.7 × *D* × *A*_430nm_(2)
where *D* is the dilution factor (if necessary, the mixture should be diluted). The relationship between the absorbance at 700 nm and 430 nm of the same sample was calculated as a validation strategy for the results; if it was lower than 0.039, it indicated that no disturbance due to suspended particles occurred.

A 2020we portable turbidimeter (LaMotte, Chestertown, MD, USA) was used for turbidity measurement, with distilled water as a blank. The same cell was used for all measurements. The Protocol for Sensory and Organoleptic Evaluation of Colombian Beers was followed for the necessary conditioning for the sample according to the type of turbidity measured [39]. In this case, the permanent and total turbidites were measured. Then the cold turbidity was estimated by following Equation (3).
(3)Cold Turbidity=Total Turbidity−Permanent Turbidity

*HPLC*: Samples of 2 mL were taken during fermentation and stored at −80 °C in Eppendorf tubes. Then the tubes were thawed and their contents were diluted to avoid possible damage to the column caused by high concentrations. Subsequently, the diluted samples were filtered through a 0.22 μm membrane filter (Hawach Scientific, Nylon, Shaanxi, China) and transferred to HPLC vials. The HPX–87H column (Bio–Rad, Hercules, CA, USA) was used to quantify carbohydrates in solutions, organic acids, short–chain fatty acids, alcohols, ketones, and neutral metabolic products [40]. The method conditions were 5 mM sulfuric acid mobile phase, 30 °C, 0.5 mL/min flow rate, an injection volume of 20 μL, and UV detection at 214 nm. Calibration curves or standards were developed for the main sugars present in the wort, the representative organic acids, and ethanol. In addition, they were tested individually and combined to verify possible peak overlapping or drifting by interactions between the sample components.

*Electronic nose*: the proposed protocol for the sensory and organoleptic evaluation of Colombian beers was followed [39]. The equipment used was the electronic nose PEN3, which consists of an array of 10 hot metal oxide sensors (Table 2). The results were combined to create aroma profiles differentiated by typology, which are recommended for identification [41].

A 20 μL sample was placed in a vial and incubated at room temperature for 1 h before taking the measurements. Twenty–five characteristic aromas present in beer were used as standard, including diacetyl, DMS, capric acid, butyric acid, acetic acid, lactic acid, isovaleric acid, butter, almond, vanilla, spiced, metallic, isoamyl acetate, indolic, geraniol, contamination smell, ethyl hexanoate, ethyl acetate, acetaldehyde, paper smell, mercaptan, oxidized, earthy, and grains or cereals.

*Electronic tongue*: The sample was filtered for the electronic tongue analysis, and then its pH was adjusted between 4 and 5 to avoid sensor damage. The equipment used was the electronic tongue Sensing System TS–5000Z (Insent, Atsugi, Japan), with the sensors AAE (Umami), CT0 (salinity), CA0 (heartburn), C00 (bitterness), AE1 (astringency), and GL1 (sweet). A sample of the initial wort was used as a reference to determine the differences that could be potentially attributed to the fermentation process.

## 3. Results and Discussion

The results are shown in two main sections. The first one depicts the encapsulates’ characterization before and after fermentation, while the second one is dedicated to analyzing the liquid phases of fermentations, including those of repeated batches with reused encapsulates.

As shown in Figure 1, the encapsulates obtained using the six proposed formulations will henceforth be referred to as C–1.a, C–1.b, C–1.c, C–2.a, C–2.b, and C–2.c. According to the results shown below, the C–2.b formulation was selected for further use in the fermentation process. These encapsulates will be called Initial Capsules (IC), followed by the name of the encapsulated strain (i.e., IC–BE–134, IC–K–97, IC–US–05, and IC–CW–12). All the selected encapsulates were used for fermentation once, except for IC–US–05, which was reused twice. These will be named FC–1 (for one repeated batch fermentation), FC–2 (for two repeated batch fermentations), and FC–3 (for three repeated batch fermentations).

### 3.1. Encapsulate Results

The diameter of the capsules varied between 3 and 3.5 mm, with those prepared with 3 percent alginate being the largest ones. Table 3 summarizes the diameters of encapsulates obtained for each formulation. In a previous study by Yadav et al., capsules with diameters ranging from 2 to 5 mm were evaluated for fermentative processes; however, those with a diameter of 3.5 mm provided the best performance [24]. The encapsulates’ size highly depends on the alginate concentration, needle diameter, and flow rate. Consequently, a wide range of encapsulate sizes can be obtained by this method.

Formulations C–1.a, C–1.b, and C–1.c result in elliptical topologies, while the other formulations exhibit a spherical shape and larger diameters (Appendix A). This is likely due to the inhomogeneity of the alginate–yeast mixture as the cell concentration increased, which caused a more significant accumulation of matter at the needle’s edge before the dripping process started. Additionally, the humidity percentage of the capsules was about the same for all formulations, with a mean of 96.6%.

#### 3.1.1. Swelling Percentage

Three samples of each formulation were submerged for 188 h in the wort to determine the swelling percentage. The changes in weight over this period were recorded at 0 h, 24 h, 44 h, 92 h, and 188 h. The swelling percentage was then determined using Equation (1) and plotted for each sample (Figure 3).

As shown in Figure 3, all encapsulates reached swelling percentages over 30 percent after 24 h, with a maximum after about 92 h. Encapsulates made with 3 percent alginate showed less swelling than those made with 1.5 percent. However, these apparent differences were not significant. Even so, 1.5 percent alginate encapsulates showed a drop in swelling from around 8 to 15 percent after 92 h. Structural changes induced by the media may have caused this behavior. Further evidence of this was provided by the macroscopic alterations observed in the encapsulates’ surface, such as the small protuberances and microfractures shown in Appendix A. Similar observations have been made previously for alginate gels [42].

#### 3.1.2. Spectroscopic Analysis

The FT–IR enabled the identification of functional groups in the sample by determining their particular vibrational bands at different wavelengths [43]. Table 4 provides a summary of the wavenumbers and their corresponding chemical bonds and functional groups. After searching in the equipment’s library for similarities with known compounds, pure alginate had the highest matching percentage.

The bands corresponding to alginate’s spectrum are shown in pink in Figure 4a. According to Table 4, the peaks in the region between 1000 and 1100 cm^−1^ exhibited C−C, C−O, and C−OH bonds characteristic of the pyranose form of the monomers present in the alginate chains [44].

In the region between 1580 and 1590 cm^−1^, the peaks are caused by symmetric stretching of COO^−^ bonds, while near 1410 cm^−1^, they are caused by asymmetric stretching of COO^−^ bonds. This is characteristic of both monomers composing the alginate chain [43,44]. Furthermore, vibrations in the region between 1480 and 1700 cm^−1^ can be ascribed to the presence of proteins in the capsule, which, in turn, are most likely originated from the yeast cells [45].

As seen in Figure 4b, the FT-IR spectra of the encapsulates of each strain after fermentation show marked differences. According to the Thermo Scientific OMNICTM software, these variations can be attributed to various compounds produced during fermentation. For example, some of them include aliphatic esters (R–COOR, region between 1060 and 1175 cm^−1^), aromatic sulfoxides (R–S(=O) −R, region between 1010 and 1000 cm^−1^), tertiary aliphatic amines (the region between 1090 and 1120 cm^−1^), aromatic chloride compounds (Ar–Cl, region between 1075 and 1110 cm^−1^), aliphatic amino acids (regions between 1580 and 1630 cm^−1^ and between 2500 and 3200 cm^−1^), aromatic amino acids (regions between 1480 and 1510 cm^−1^, between 1550 and 1650 cm^−1^, and between 2600 and 3100 cm^−1^), aromatic ethers (Ar−O−Ar, between 1210 and 1250 cm^−1^), and primary aliphatic alcohols (R−CH_2_−OH, regions between 1000 and 1150 and between 3200 and 3600 cm^−1^).

Some post-fermentation peaks were detected in the 1700 to 1480 cm^−1^ wavenumber region, which likely corresponded to peptide bonds that can be correlated to the presence of C=O, C−N, and N-H functional groups. In addition, aliphatic and aromatic amino acids derived from wort were detected between 2800 and 3000 cm^−1^ [45]. Finally, in the region between 3200 and 3500 cm^−1^, the presence of primary aliphatic alcohols is most likely related to the fermentative process.

#### 3.1.3. Mechanical Resistance

A series of tests were conducted to compare the breaking force and elasticity of different types of encapsulates. In such tests, the sample was pierced with a 2 mm needle system (P2). Appendix A provides graphical details of these tests.

A Tukey test confirmed, as shown in Figure 5, that there is a significant difference between the required breaking forces for C−2.b and C−2.c encapsulates and the other samples. This is consistent with a previous report for capsules containing 3 percent alginate in which the values ranged from 3 to 25 N [46]. Additionally, it is possible to assert that there are statistically significant differences in elasticity due to the different alginate concentrations. However, the analysis also revealed that varying calcium chloride concentrations did not impact elasticity.

Three batches of reused encapsulates were subjected to the same test to identify any changes that may have occurred during the repeated batch reuse. Comparing the IC–US-05 breaking force to that of encapsulated FC–1, FC–2, and FC–3 in Figure 5c, it is evident that this value decreases significantly after fermentation. Previous research on alginate encapsulates indicates that sugars such as sucrose can significantly reduce the breaking force values [46]. Therefore, it is highly probable that the presence of sugars in the wort, such as maltose, glucose, or fructose, could reduce resistance following fermentation. Due to the low resistance of the samples, it was also impossible to measure their elasticity after fermentation.

#### 3.1.4. Thermal Stability Analysis

A thermogravimetric analysis (TGA) was performed on a 48-h lyophilized sample of each encapsulate formulation by raising the temperature to 600 °C. However, no changes were observed in the thermograms after 250 °C. Figure 6 summarizes the TGA thermograms for each of the six formulations. The initial weight loss for all samples occurred prior to 100 °C and is attributable to water evaporation. These losses are 49.5% for the 1.5 percent alginate capsules and 41.8% for the 3 percent alginate capsules (Figure 6a,b), which may indicate that even after 48 h of lyophilization, some water remains in the samples, so the lyophilization time could have increased. In addition, the TG curves for 0.05 and 0.1 M CaCl_2_ exhibited similar behavior for both alginate concentrations, resulting in stabilization at about 30% for the 1.5 percent alginate formulation and at about 17% for the 3 percent alginate formulation. Figure 6c,d illustrate similar inflection points for the three calcium chloride concentrations, which can be associated with water vaporization.

The FC–1, FC–2, and FC–3 encapsulates underwent additional TGA analyses (Figure 7). Compared to the initial encapsulates, where approximately 41.5% of the weight was lost upon reaching 100 °C and a stable point of 17.5 weight percent at 250 °C, the weight loss after two repeated batches is between 23.7 and 26.6% (Figure 7b,c). For the third repeated batch, the initial weight loss decreased to approximately 6.1% (Figure 7d). It stabilized at 82.58%, indicating that the capsule was formed not only by the alginate but also by other adsorbed components.

The DTG profile indicates weight losses at 150 and 200 °C for repeated batches 1 and 2. (Figure 7b,c). On the encapsulates, the caramelization of residual sugars and the deposition of other substances from the wort can cause such changes. In the third repeated batch, there are two additional and distinguishable minor weight losses at temperatures between 160 and 180 °C. Lastly, there is a significant weight loss for all samples heated above 250 °C, which correlates with the decomposition of alginate backbone chains [31]. In conclusion, the behavior of TG profiles may be linked to a multi–stage breakdown of unstable components originating from wort and fermentation. Identifying these compounds, which will be the subject of future research, was not possible [47].

#### 3.1.5. Morphology

To examine the microstructure of the encapsulates, SEM imaging was conducted for their surface, edge, and center. Such images were analyzed with the ImageJ^®^ software to determine the pore size distribution. Figure 8 and Figure 9 show, from left to right, a general encapsulate image, the encapsulates’ center and their edge. Some encapsulated yeast cells are visible in the images, denoted by yellow circles. The 1.5% alginate encapsulates (Figure 8) have a disordered pore structure, with smaller pores at the center and larger pores near the edges. The C–1.c formulation shows encapsulates with a more compact pore structure composed of pilled plaques and a dense lattice at the center.

In contrast, the alginate encapsulates containing 3 percent alginate (Figure 9) have a completely different pore structure. Even though the C–2.a formulation is still disordered, its pores are more defined than those of alginate at a concentration of 1.5 percent. The C–2.b formulation has an ordered structure with well–defined hexagonal pores throughout the capsule, which might facilitate the mass transport of solutes entering and leaving the capsule, which is an essential exchange for fermentation processes. Lastly, the C–2.c formulation displays ordered elongated sheets that fold inward from the capsule’s center to its periphery.

The FC–1, FC–2, and FC–3 encapsulates from repeated batches were also examined via scanning electron microscope. Figure 10 shows, from left to right, micrographs of the surface, the center, and the edge of each encapsulate. The FC–1 encapsulates exhibit some surface protuberances, but the external pores can still be observed. The center of the encapsulate contains a structured array of pores enclosing yeast cells that proliferated during fermentation. FC–2 encapsulates exhibit a more degraded and less stable structure than FC–1 encapsulates; however, a high cell density can still be observed. The FC–3 encapsulates display apparent signs of structural degradation without affecting cell density negatively.

The pore size values of the encapsulates displayed a high level of variation and marked deviations with respect to each other (Figure 11). In addition, the porosity of the 3 percent alginate encapsulates was higher, with the C–2.b formulation having the highest. However, it was not significantly different from the other encapsulates. These observations can be caused by diverse factors, like the disposition of the Ca ions in each formulation, the time required to form spheres [23], and the presence of yeast and contaminants such as some emulsifiers coming from the commercial strains [48].

Although the FC–1, FC–2, and FC–3 encapsulates exhibit a decrease in pore size, such reduction was not statistically significant (*p* > 0.05). This can be attributed to the growth and accumulation of yeast, which largely impede the collection of accurate values.

In addition, the diffusivity of the substrate and yeast was evaluated for each formulation to estimate the possible impact of structural differences on the exchange of nutrients and residues to and from the encapsulates during fermentation. The diffusivity was determined by adjusting a multiphysics model, presented in Appendix A, to the experimental results of methylene blue diffusion, which is used as a model molecule, as shown in Appendix A. The results of this diffusivity test are presented in Appendix A [49,50].

#### 3.1.6. Viability

The change in yeast viability in the encapsulates was tracked from its initial elaboration to its third reuse after repeated batch fermentations. For this stage, the selected strain was the US–05, mainly because of its neutral profile, which allows the isolation of the impact of the yeast’s presence in the sensory profile. Hence, only the IC–US–05, FC–1, FC–2, and FC–3 encapsulates were evaluated for this test. Figure 12 shows the confocal images analyzed with the Fiji–ImageJ^®^ software, in which live yeast cells are colored red throughout the entire population (blue cells).

Initial IC-US05 encapsulates reached 93.74% viability. With each fermentation, however, this value diminished. The viability for FC–1 encapsulates was 88.06%, while that of FC–2 encapsulates was 38.80%. In the case of FC–3 encapsulates, it reached 32.50%. Depending on the yeast strain, this loss of viability may be caused by various factors, including osmotic stress or exposure to highly alcoholic environments [51].

### 3.2. Fermentation Results

The 1.5% alginate capsules showed fractures on their surface and structural changes during the swelling test, which was conducted by immersing them in the culture medium (wort) for a few days. As a result, they were disregarded for use in the fermentation stage of the study. For the 3% alginate capsules, the selection of the C–2b formulation was made based on the results for breaking force and morphology.

#### 3.2.1. Yeast Growth

A growth curve was built for each strain, as shown in Figure 13. The American strain had the lowest lag time (about 1 h), while the Belgian, German, and Colombian strains required between 2 and 3 h to reach the exponential phase. All the strains reached the stable phase after approximately 9 h of growth. The specific growth rate (µ) for each strain was determined from the slope of the curves during the exponential phase. µ for the BE–134 strain was 0.240 h^−1^, which was similar to the value of 0.249 h^−1^ for the K–97 strain. For the US–05 strain, µ approach 0.199 h^−1^, while for the CW–12 strain, it was 0.195 h^−1^.

The progress of the fermentation was monitored for 14 days using a refractometer to measure the Brix degrees (°Bx). Notably, the results obtained with this instrument may vary considerably due to the reliance on the individual’s visual acuity.

The equivalent values for Plato degrees (°P) after converting from the °Bx obtained during fermentation are plotted in Appendix A. Slight differences of less than 0.5 °P were observable between fermentations. However, they are still statistically significant in the case of encapsulated vs. free yeasts for the K–97 and US–05 strains (*p* < 0.05). This could be due to metabolic disturbances in immobilized cells, which can speed up substrate consumption and ethanol production. However, it has been found that it also reduces the production of secondary metabolites [52]. A noticeable difference of 2 °P (*p* < 0.05) is observed for the Colombian strain (Appendix A), which indicates that sugar consumption might increase upon encapsulation. Due to its wild nature, this strain has not been adapted for increased resistance to alcohol or other environmental factors, resulting in diminished performance [52,53]. This may explain why it seems to perform better when encapsulated.

For repeated batch experiments, the °P values were very similar (*p* > 0.05), suggesting that the process can be completed successfully, albeit with different organoleptic profiles, despite the reduced viability after each reuse cycle (Appendix A). In addition to measuring °P, spectrophotometric monitoring of yeast growth was performed throughout the fermentation process. There was a difference in the amount of biomass, possibly because the in–capsule biomass was not measured. After fermentation, the maturation process was carried out to determine the final organoleptic profile.

#### 3.2.2. Color and Turbidity

The malts used to brew largely determine the beer’s color, which is one of the most important quality indicators. However, it has been reported that time and storage conditions also affect this property. Due to the oxidation of catechins in malt, color alterations are possible [54,55]. For all the fermentations, the selected malt was Pilsner, which gives a blonde hue to the beer as the Maillard reactions take place while preparing the wort. Some phenolic compounds may also interfere [55] and cause a color value of around 4.45 ± 0.65 on the SRM scale (8.77 EBC). The evaluated fermentations exhibit negligible color differences that are not visible to the naked eye (Table 5). The Dunnett and Tukey tests allow us to conclude that these differences are not statistically significant (*p* > 0.05).

Turbidity is an additional important quality parameter, and its value tends to be greater for craft beers, though it can vary depending on the manufacturing process and beer style. Some beers on the market have turbidities of less than 10 NTU, whereas others can approach 230 NTU [56]. The total turbidity of the wort was 216.5 ± 0.7 NTU, while the permanent turbidity was 5.4 ± 0.5. The results for the turbidity of the other fermentations are shown in Figure 14.

Except for the US–05 and its repeated batches, this study found a statistically significant difference (*p* < 0.05) in total turbidity between beers made with encapsulated and free yeasts. This difference may be explained by the fact that the K–97 and Belgian strains are low–flocculation yeasts. Therefore, they tend to stay suspended in the medium longer than the American and Colombian strains. Moreover, it has been reported that calcium ions may induce yeast flocculation, resulting in more translucent beers [57].

#### 3.2.3. HPLC

Liquid chromatography was used to evaluate the production of ethanol and organic acids and the consumption of the substrate. The corresponding concentration profiles are presented in Figure 15. It was observed that for the commercial strains, ethanol production was similar for the encapsulated and free yeast (*p* > 0.05). In contrast, for the Colombian wild strain, the ethanol production was significantly enhanced when encapsulated (*p* < 0.05). Furthermore, substrate consumption for the wild strain was up to 10 times higher compared with encapsulated yeast.

For the fermentations performed with the US–05 strain, the final concentration of ethanol was 6.42 ± 0.24 (%*v*/*v*) for the initial capsules, while it was 7.13 ± 0.05 and 6.43 ± 0.38 for the second and third repeated batches, respectively. These differences were not significant. This could imply that, despite the decrease in cell viability, performance was not affected detrimentally. Furthermore, the proliferation of yeast inside the encapsulate may have allowed comparable substrate consumption levels and ethanol production to those of the initial batch. Such proliferation results are shown in Figure 10.

Additionally, most likely because of the protection provided by the capsule, the CW–12 strain showed an increase in alcohol production of 1.23 mg/mL [42]. The results obtained for the acetic and lactic acids on the 14th day for all the fermentations are summarized in Table 6. The initial wort showed a pH value of 5.44 ± 0.23 and concentrations of 8.96 ± 0.89 and 0.49 ± 0.07 mg/mL for lactic and acetic acids, respectively. In the case of the reused encapsulates FC–1 and FC–2, the production of organic acids increased, which can explain the reduction in pH compared with the initial IC–US–05 encapsulates. However, such increments in the production of acids are not statistically significant between batches.

The K–97 showed a significant increase in lactic acid production of 6.35 mg/mL after encapsulation, indicating that K–97 is highly sensitive to the encapsulation process. All the other strains showed no significant differences after encapsulation.

The CW–12 strain showed a marked difference in the production of organic acids when compared to the other strains. In particular, CW–12 showed the highest production of acetic acid regardless of whether it was free or encapsulated. This can be explained by its previously reported high tolerance to acetic acid, as evidenced by a production of up to 37.6 ppm in only 36 h [30]. Fermentation time and culture conditions might also be responsible for an increase in the organic acid production rate. Additionally, after each repeated batch, there was a non–statistically significant increase trend in the production of acids.

#### 3.2.4. Electronic Nose

The electronic nose was used to compare 21 distinct standard beer aromas; however, only a handful were identified in each sample. The aromatic profiles of each strain are shown in Figure 16 and Figure 17. In the case of the Belgian strain, peppery and spicy aromas predominate, with ethyl acetate (the primary ester in beers and wines), ethyl hexanoate, acetaldehyde, and grainy notes. Fermentations with the free BE–134 strain increase capric acid concentration and a lack of isoamyl acetate; however, these trends are reversed when encapsulates are used.

The predominant aromas of the K–97 strain are peppery and spicy due to the presence of ethyl acetate and caprylic acid. However, the encapsulates show a trend of aroma intensification with no statistical significance. In addition, isoamyl acetate was produced, and the ethyl hexanoate that was likely initially present in the wort was retained by the free yeast.

The CW–12 profile is dominated by ethyl acetate, with hints of ethyl hexanoate. Using encapsulates produces earthy and hefeweizen aromas. Additionally, the intensity of capric acid and the pungent odor decreased while that of acetaldehyde increased. The US–05 strain’s encapsulate profile differed the most with respect to the other evaluated strains. In this instance, aromas of DMS, almond, hefeweizen, and acetaldehyde are produced. Additionally, the intensity of capric acid, the pungent odor, and ethyl acetate showed a slight decrease.

Yonezawa and Fushiki present information regarding the primary compounds that impart flavor and aroma to beer, including those produced during fermentation and maturation, such as ketones, aldehydes, esters, organic acids, among many others. Isobutyl acetate (banana, sweet, fruity), isoamyl acetate (banana), ethyl acetate (fruity, sweet), and acetaldehyde (apple, green leaves) can be found among ketones and aldehydes. Other short–chain aldehydes are associated with foul odors. Some by–products, such as glycerol, may be present in low concentrations, but we failed to relate them to any identifiable aromas [10].

In the presence of sulfur compounds, substances such as DMS (egg odor) and mercaptans (some of which are related to unpleasant odors like decomposed food) can be generated. Certain organic acids, including butyric, isovaleric, and capric, emit rancid odors reminiscent of cheese or butter that are undesirable in beer. The acetic acid imparts a vinegar aroma to the product, and these acids are responsible for beer’s astringency [10].

Figure 18 shows the profiles of the reused encapsulates (i.e., after repeated batches). According to the results, capric acid decreased with each repeated batch, while aromas such as paper, grainy, hefeweizen, and spicy emerged after the second repeated batch. Vanilla and acetaldehyde aromas appeared after the third repeated batch, while others, such as paper, diminished. After repeated batch experiments, the ethyl acetate odor was always present and predominated. Variations in the aromatic profile of these encapsulates may be attributable to the loss of viability and potential genetic alterations in the microorganisms, which can result in the production of undesirable metabolites [57].

#### 3.2.5. Electronic Tongue

Five measuring cycles were performed with the electronic tongue for each sample. However, only the first option was considered to avoid the potential negative impact of prolonged exposure to air and light on the profile. According to recent reports, these two factors cause variations in the sensory profile by generating undesirable substances. Figure 19 displays the normalized flavor values relative to the fermentations with free yeast for each strain. The sourness and richness parameters showed the least variation; however, the sourness may have been impacted by the pH adjustment necessary to prevent sensor damage. There were no significant differences (*p* > 0.05) in the evaluated flavors between free and encapsulated CW–12 yeasts (Figure 19c). Some minor variations were observed in the bitterness, with K–97 encapsulated yeast producing a slightly lower bitterness than the other strains (Figure 19b). This might be attributed to the presence of catechins and polyphenols [56].

As shown in Figure 19a,b, the astringency decreased for encapsulated Belgian and German strains. The same was true for the aftertaste. It has been reported that the production of organic acids is associated with astringency [10]. This is consistent with the concentrations of lactic and acetic acids evidenced in the HPLC results for these encapsulates. The sweetness of the K–97 and US–05 encapsulates decreased compared with the free strains. This is consistent with the HPLC results for the German strain, according to which residual sugar concentrations were below those made with free yeast.

Saltiness diminished upon encapsulation, possibly due to sodium adsorption on the alginate. Umami was reduced slightly for encapsulated K–97 yeast and even more noticeably for US–05 encapsulates, which may indicate a decrease in monosodium glutamate production.

Despite the increased amount of residual sugars and organic acid production, in the fermentations with FC–1 and FC–2 encapsulates (Figure 20), salty flavor, richness, and aftertaste increased, while astringency and sweetness decreased.

## 4. Conclusions

An emerging market trend in the food and beverage industry is that consumers are constantly searching for premium products of high–quality, with potential health benefits and pleasant taste. This can be applied to the case of beer, where consumers are looking for more complex flavor profiles. A route to alter the sensory profiles of food and beverages is the encapsulation of compounds with unique flavors. Here, we employed this approach to develop a strategy to alter the sensory profile of beer by brewing with alginate encapsulates of yeast. The selected formulation for encapsulation was 3% (*w*/*v*) sodium alginate and 0.1 M calcium chloride, mainly because of its ordered and hexagonal porous structure, which allows the unimpeded exchange of metabolites, and its high breaking force (12N), which helps to preserve material integrity at an acceptable level during fermentation, and even after.

We conducted three repeated batches to evaluate the potential reusability of the encapsulates. The viability of the yeast decreased by 5.68% from the initial encapsulates to the first repeated batch and by 61.24% to the third one. However, the ethanol production remained at a high level and only decreased by around 0.5% between fermentations.

We identified differences in the sensory profiles obtained between encapsulated and free yeast; however, such variations were inconsistent between strains. While astringency and saltiness decreased for the BE–134 strain, the K–97 strain profile varied in almost every aspect except for richness. On the other hand, the wild CW–12 strain showed almost no changes between profiles, which suggests that this approach might be useful to operate with yeast strains with low adaptation to alcoholic fermentation environments without sacrificing yield. This is the subject of future research with an expanded arsenal of locally isolated wild strains.

Our results indicate that encapsulation of yeast in alginate appears to be a viable and promising option to alter sensory profiles of beer, while maintaining a superior alcohol production yield.

Further research is essential to explore capsule production of a controllable and homogeneous size as this might have a significant impact on the profiles. This could be enabled by microfluidic platforms. Additionally, we plan to research potential coatings for other co–polymers to increase yeast viability and extend the life of the capsules. In terms of obtaining more complex profiles, we are currently exploring the possibility of adding encapsulated hops to the wort.

## Figures and Tables

**Figure 1 polymers-15-01742-f001:**
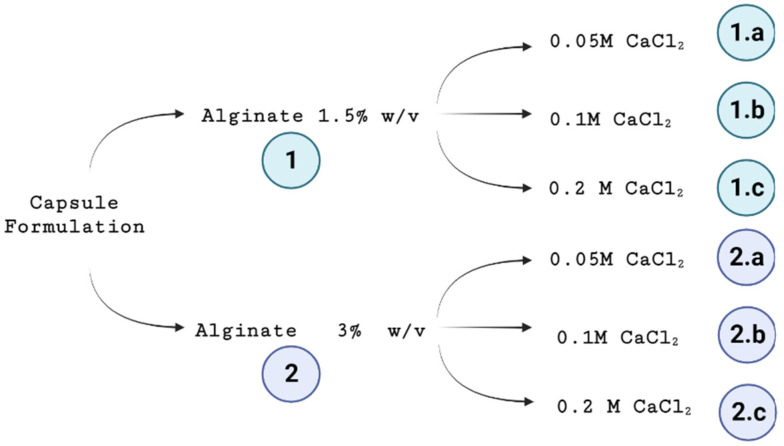
Experimental design for the calcium alginate–yeast encapsulate formulation.

**Figure 2 polymers-15-01742-f002:**
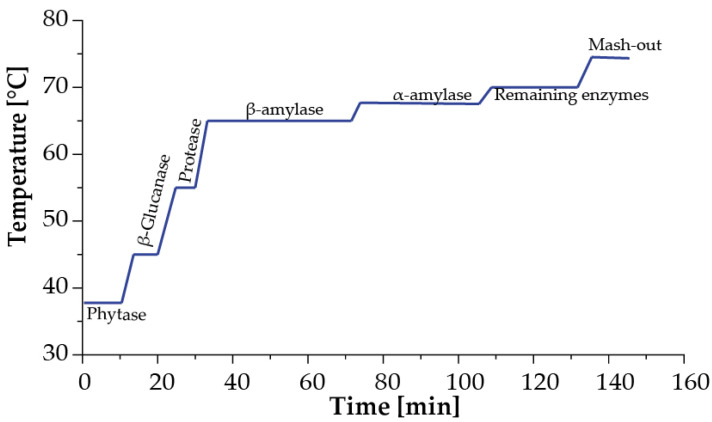
Temperature profile for the mashing process. The enzymatic activation takes place sequentially in the following order: 1. phytase (10 min at 38 °C); 2. β–Glucanase (10 min at 45 °C); 3. protease (10 min at 55 °C); 4. β–amylase (45 min at 65 °C); 5. α–amylase (30 min at 68 °C); 6. remaining enzymes (30 min at 70 °C). Finally, enzymes are inactivated in a mashup stage (10 min at 75 °C).

**Figure 3 polymers-15-01742-f003:**
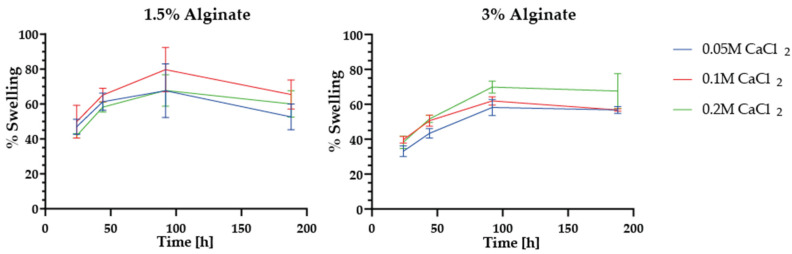
Swelling percentage profiles of the encapsulates were obtained after 188 h in wort for 1.5% and 3% alginate contents and for all three concentrations of calcium chloride.

**Figure 4 polymers-15-01742-f004:**
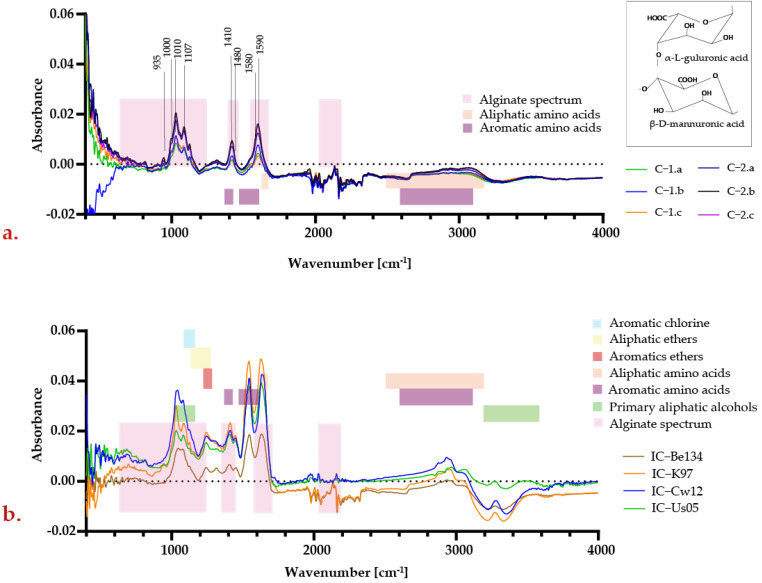
FT-IR spectra for yeast–calcium alginate encapsulates: (**a**) spectra for the initial encapsulates of all the formulations; (**b**) spectra for the after-fermentation capsules of all the strains.

**Figure 5 polymers-15-01742-f005:**
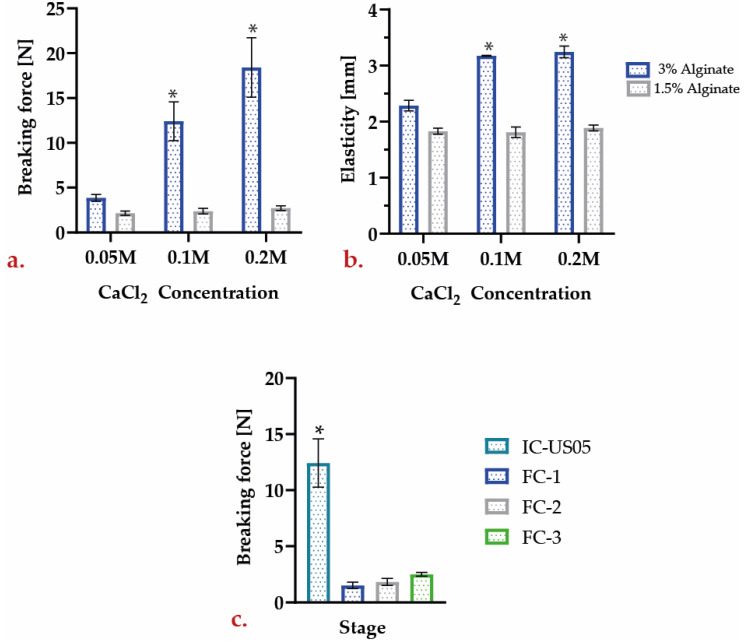
Mechanical properties for initial capsules: (**a**) breaking force; (**b**) elasticity; (**c**) reduction of the breaking force of encapsulates used in repeated batches (* Significantly different sample when compared between them).

**Figure 6 polymers-15-01742-f006:**
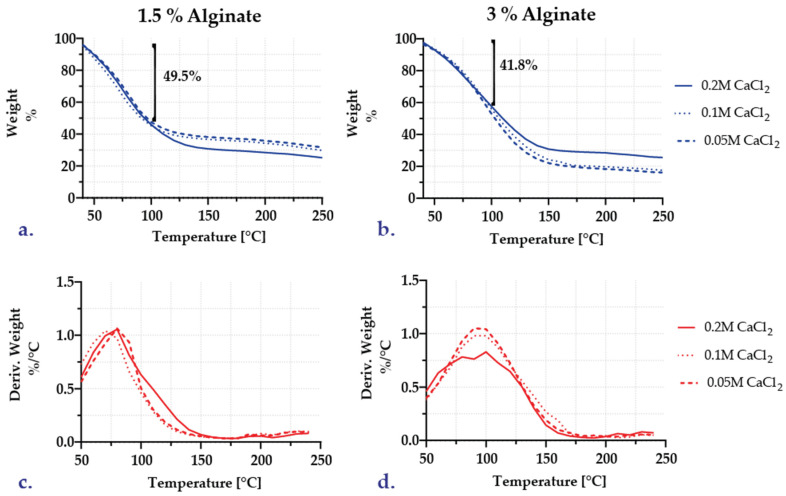
Thermograms for encapsulates: (**a**) TG curves for 1.5% alginate capsules; (**b**) TG curves for 3% alginate capsules; (**c**) DTG curves for 1.5% alginate capsules; (**d**) DTG curves for 3% alginate capsules.

**Figure 7 polymers-15-01742-f007:**
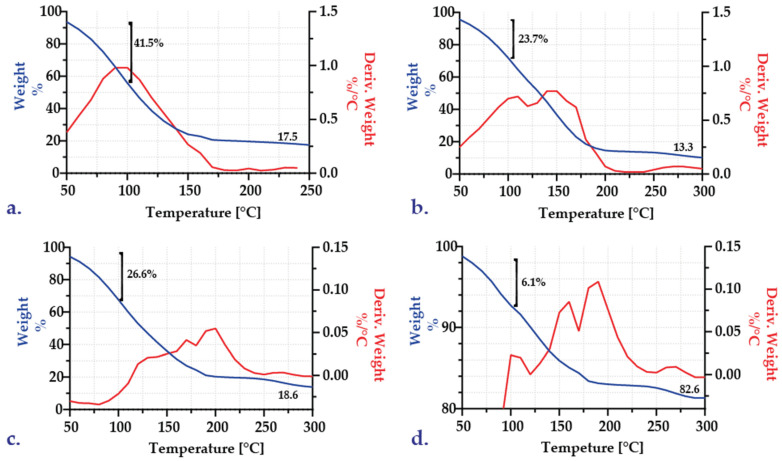
Comparative thermograms for the encapsulates post–fermentation in repeated batches: (**a**) initial capsules IC–US–05; (**b**) encapsulates after one repeated batch (FC–1); (**c**) encapsulates after two repeated batches (FC–2); (**d**) encapsulates after three repeated batches (FC–3).

**Figure 8 polymers-15-01742-f008:**
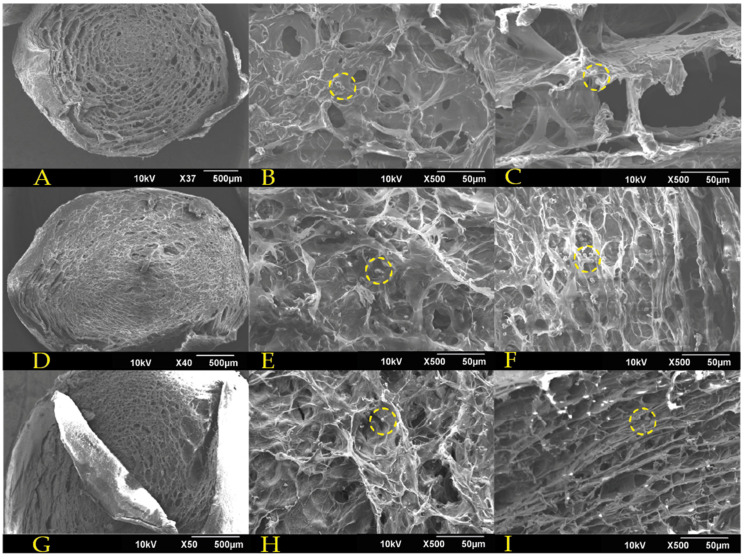
SEM images for 1.5% alginate encapsulates: (**A**) general view of C–1.a formulation; (**B**) encapsulate center of C–1.a formulation; (**C**) encapsulate edge of C–1.a formulation; (**D**) general view of C–1.b formulation; (**E**) encapsulate center of C–1.b formulation; (**F**) encapsulate edge of C–1.b formulation; (**G**) general view of C–1.c formulation; (**H**) encapsulate center of C–1.c formulation; (**I**) encapsulate edge of C–1.c formulation. Yeast cells are shown with yellow dotted circles.

**Figure 9 polymers-15-01742-f009:**
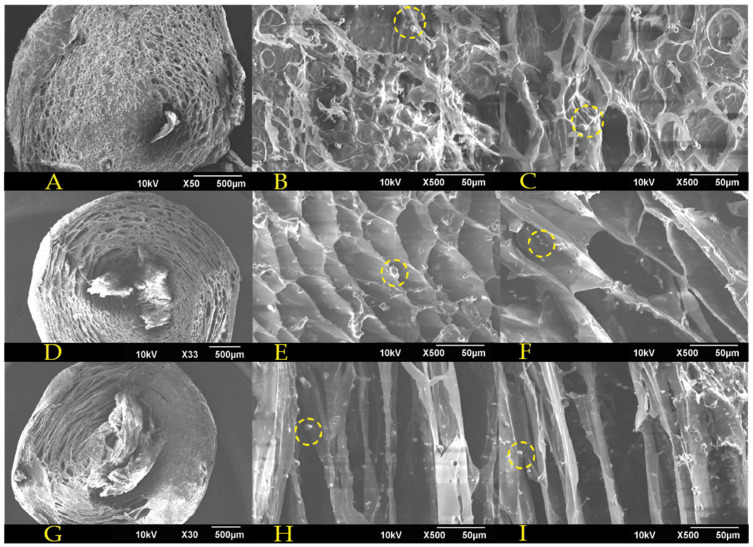
SEM images for 3% alginate encapsules: (**A**) general view of C–2.a formulation; (**B**) encapsulate center of C–2.a formulation; (**C**) encapsulate edge of C–2.a formulation; (**D**) general view of C–2.b formulation; (**E**) encapsulate center of C–2.b formulation; (**F**) encapsulate edge of C–2.b formulation; (**G**) general view of C–2.c formulation; (**H**) encapsulate center of C–2.c formulation; (**I**) capsule edge of C–2.c formulation. Yellow dashed circles represent yeast cells.

**Figure 10 polymers-15-01742-f010:**
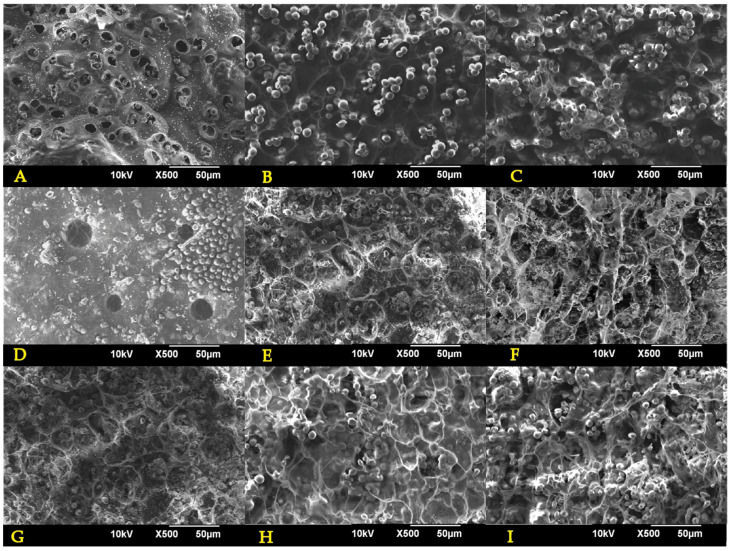
SEM images for encapsulates post–fermentation in repeated batches: (**A**) surface view of FC–1 formulation; (**B**) encapsulate center of FC–1 formulation; (**C**) encapsulate edge of FC–1 formulation; (**D**) surface view of FC–2 formulation; (**E**) encapsulate center of FC–2 formulation; (**F**) encapsulate edge of FC–2 formulation; (**G**) surface view of FC–3 formulation; (**H**) encapsulate center of FC–3 formulation; (**I**) encapsulate edge of FC–3 formulation.

**Figure 11 polymers-15-01742-f011:**
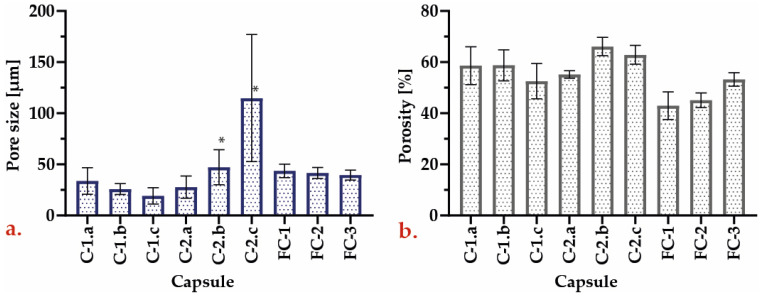
(**a**) Pore size for tested capsules (* Significantly different sample when compared between them); (**b**) porosity for tested capsules.

**Figure 12 polymers-15-01742-f012:**
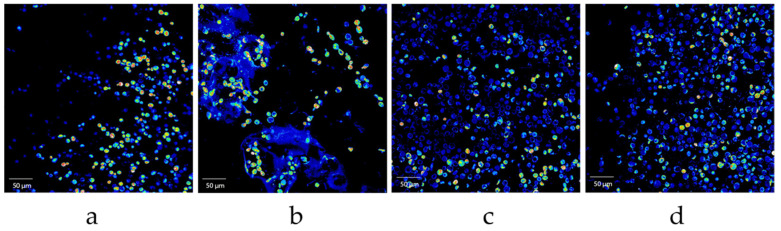
Confocal microscopy images for yeast viability 40× (Total cells: blue; live cells: red): (**a**) IC–US–05 encapsulates; (**b**) encapsulates after one repeated batch (FC–1); (**c**) encapsulates after two repeated batches (FC–2); (**d**) encapsulates after three repeated batches (FC–3).

**Figure 13 polymers-15-01742-f013:**
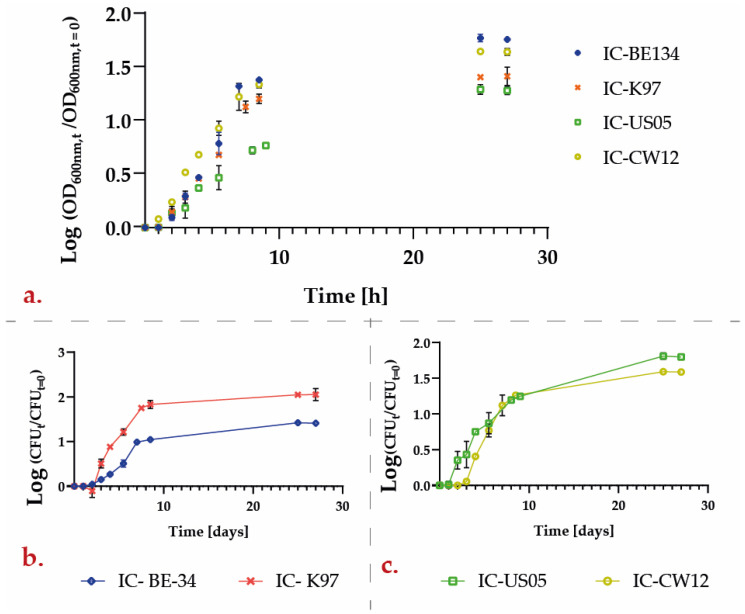
Time evolution of yeast growth. (**a**) Log(OD_600_/OD_600t=0_) for all strains; (**b**) Log(CFU_t_/CFU_t=0_) for BE–134 and K–97 strains; (**c**) Log(CFU_t_/CFU_t=0_) for US–05 and CW–12 strains.

**Figure 14 polymers-15-01742-f014:**
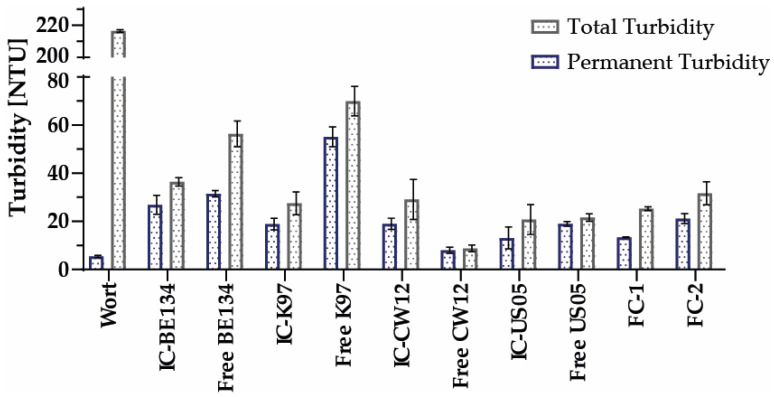
Turbidity results for the fermentations performed with encapsulated and free yeast and from various batches.

**Figure 15 polymers-15-01742-f015:**
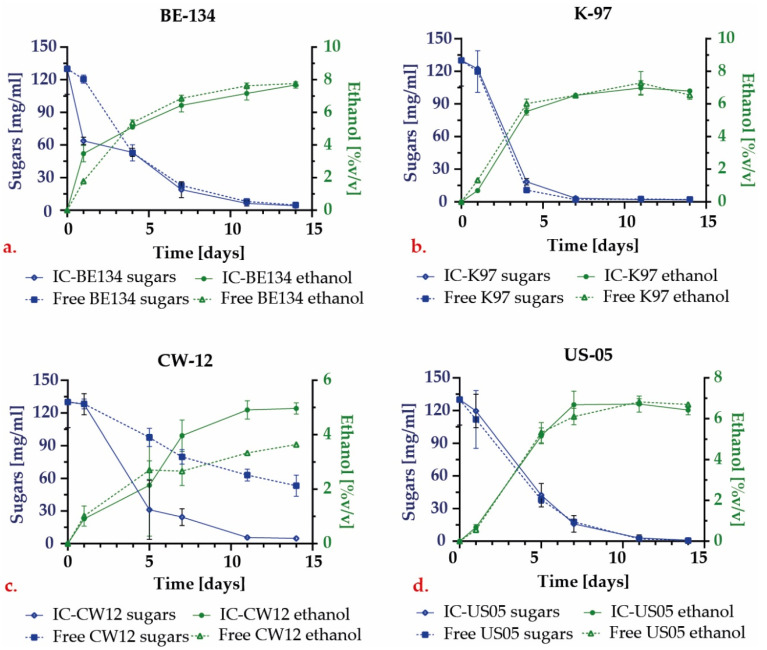
Comparative ethanol production profile and sugar consumption during fermentation with encapsulated and free yeast for (**a**) BE–134 strain, (**b**) K–97 strain, (**c**) CW–12 strain, (**d**) US–05 strain.

**Figure 16 polymers-15-01742-f016:**
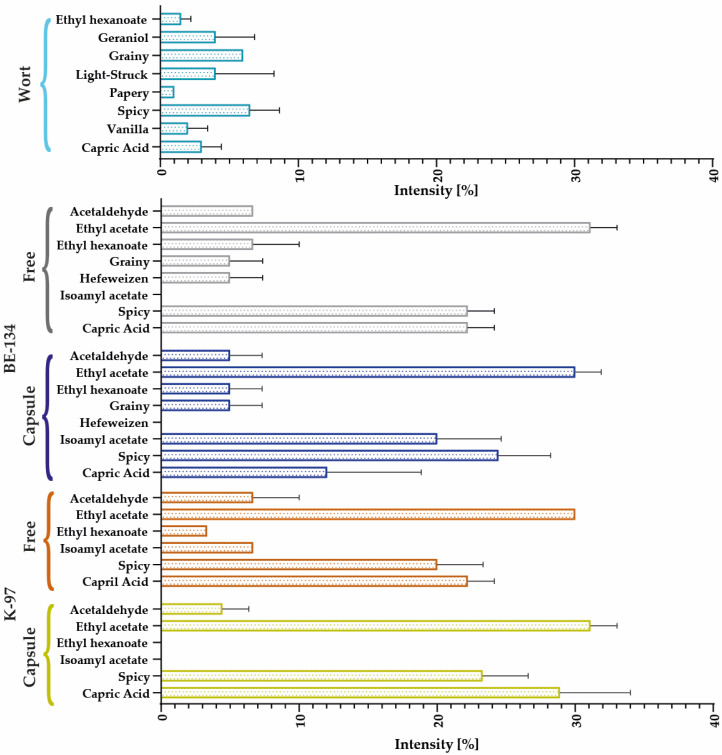
Comparative aromatic profiles of encapsulated and free yeast for the BE–134 and K–97 strains.

**Figure 17 polymers-15-01742-f017:**
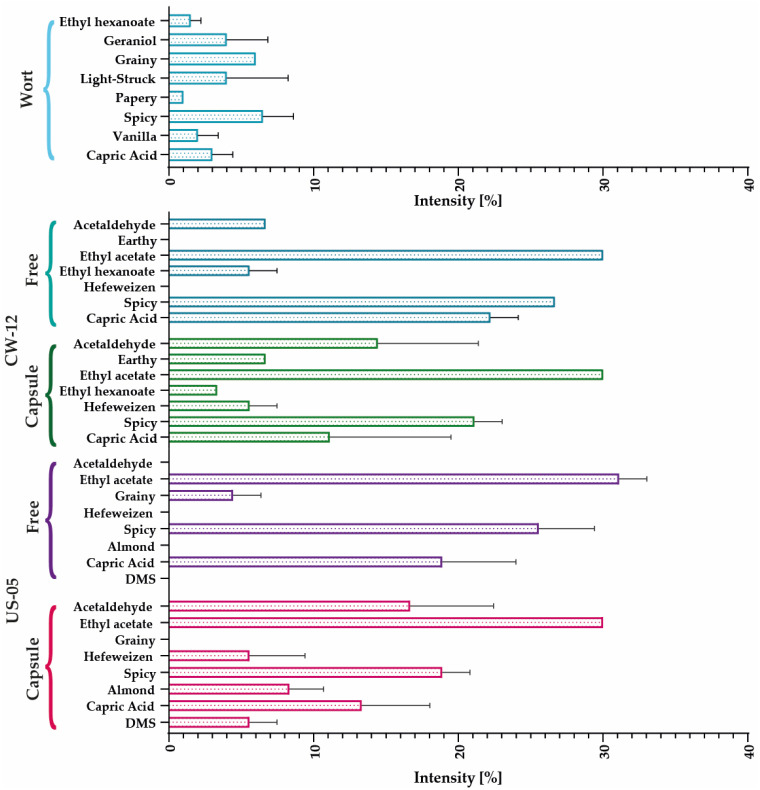
Comparative aromatic profiles of encapsulated and free yeast for the CW–12 and US–05 strains.

**Figure 18 polymers-15-01742-f018:**
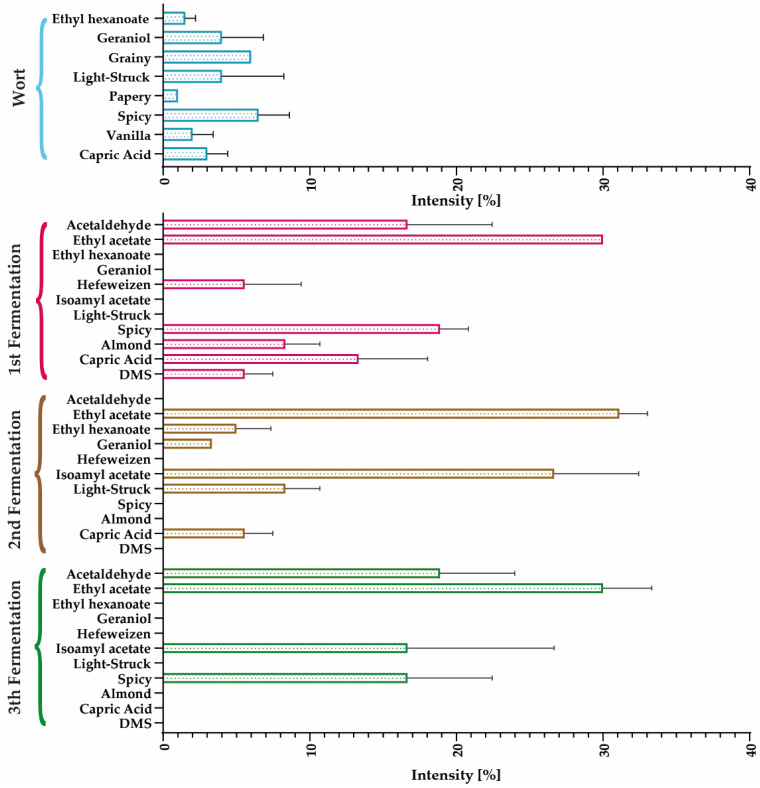
Comparative aromatic profiles of repeated batch fermentations with reused encapsulates.

**Figure 19 polymers-15-01742-f019:**
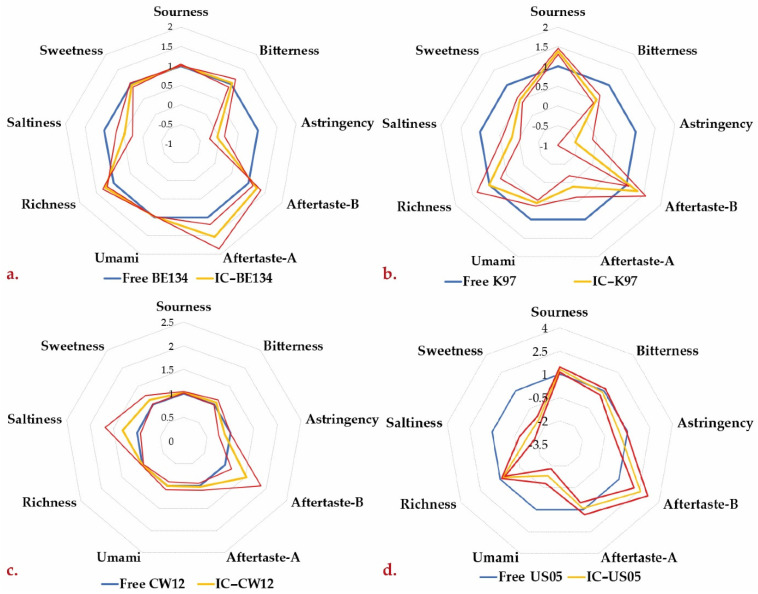
Comparative electronic tongue results for encapsulated and free yeast: (**a**) BE–134 strain; (**b**) K–97 strain; (**c**) CW–12 strain; (**d**) US–05 strain.

**Figure 20 polymers-15-01742-f020:**
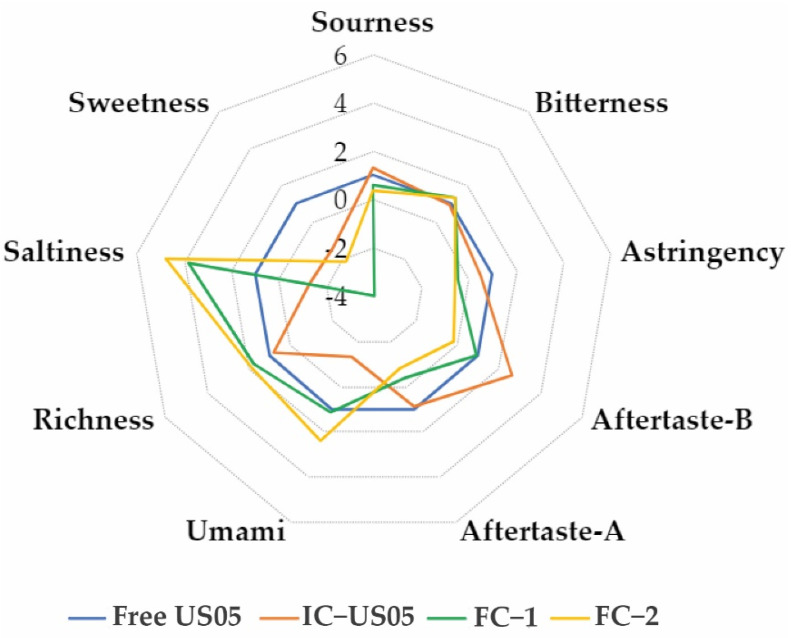
Electronic tongue results for reused encapsulates after repeated batches.

**Table 1 polymers-15-01742-t001:** *S. cerevisiae* strains used in this study.

Strain	Trade Name	Origin	Laboratory	Ref.
BE–134	SafAleTM BE–134	Belgian	Fermentis by Lesaffre	
K–97	SafAleTM K–97	German	Fermentis by Lesaffre	
US–05	SafAleTM US–05	American	Fermentis by Lesaffre	
CW–12	CW12	Colombian	Universidad de los Andes	[30]

**Table 2 polymers-15-01742-t002:** PEN3 electronic nose sensors [41].

Sensor	Related Compounds
W5S	Nitrogen oxides and ozone
W3C	Ammonia
W6H	Hydrogen component
W5C	Aromatic and aliphatic (slightly polar) compounds
W1S	Methane
W1W	Sulfur organic compounds
W2S	Partial polar alcohols and compounds
W2W	Sulfur and aromatic components
W3S	High–concentration compounds and methane

**Table 3 polymers-15-01742-t003:** Weight and diameter of the capsule formulations.

Encapsulate	Weight [mg]	Diameter [mm]
C–1.a	18.37 ± 0.23	3.05 ± 0.00
C–1.b	17.17 ± 0.64	3.25 ± 0.15
C–1.c	17.63 ± 0.29	3.10 ± 0.00
C–2.a	25.60 ± 0.30	3.50 ± 0.00
C–2.b	24.60 ± 0.53	3.50 ± 0.10
C–2.c	24.23 ± 0.23	3.43 ± 0.33

**Table 4 polymers-15-01742-t004:** FT-IR detection bands for several chemical bonds and functional groups present in the alginate spectrum.

Wavenumber (cm^−1^)	Assignment	Molecular Vibration
1700–1480	C=O, C−N	Stretching
1700–1480	N−H	Bending
1649–1410	R−COO^-^	Symmetric and asymmetric Stretching
1200–870	C−O, C−C	Stretching
1107–935	C−O	Stretching
1107–935	C−C−H, C−OH	Bending

**Table 5 polymers-15-01742-t005:** Color results for fermentations performed with encapsulated and free yeast and different batches.

Color (SRM)	BE–134	K–97	CW–12	US–05		
				Initial	FC–1	FC–2
Free	4.21 ± 0.01	4.42 ± 0.09	4.07 ± 0.37	4.19 ± 0.01	–	–
Encapsulate	3.97 ± 0.24	3.67 ± 0.02	3.47 ± 0.23	3.61 ± 0.26	4.48 ± 0.03	3.48 ± 0.58

**Table 6 polymers-15-01742-t006:** Acetic and lactic acid HPLC results for all the fermentations on the 14th day.

Organic Acid	Yeast	BE–134	K–97	CW–12	US–05		
					Initial	FC–1	FC–2
Acetic [mg/mL]	Free	0.79 ± 0.07	0.75 ± 0.21	1.77 ± 0.18 *	0.48 ± 0.26	–	–
	Encapsulate	0.74 ± 0.15	0.58 ± 0.13	1.97 ± 0.07 *	0.91 ± 1.17	1.69 ± 1.42	4.88 ± 1.78
Lactic [mg/mL]	Free	8.55 ± 0.94	11.28 ± 1.64	9.53 ± 1.52	10.77 ± 1.37	–	–
	Encapsulate	11.84 ± 5.09	17.63 ± 1.33 *	8.87 ± 0.47	7.85 ± 2.00	13.71 ± 0.54	9.87 ± 1.45
pH	Free	4.52 ± 0.19	4.04 ± 0.28	3.24 ± 0.10	4.01 ± 0.04	–	–
	Encapsulate	4.29 ± 0.52	3.62 ± 0.03	3.29 ± 0.01	3.82 ± 0.29	3.71 ± 0.02	3.57 ± 0.03

* Significantly different sample when compared between strains.

## Data Availability

Not applicable.

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
