# Peer review of "The Impact of Yeast Encapsulation in Wort Fermentation and Beer Flavor Profile"

_polymers, 2023, doi:10.3390/polym15071742_

Round 1

Reviewer 1 Report

What is the reason for pre-culturing the lyophilized strains? The easier way is to put the lyophilized culture directly into the alginate.
The specification of the alginate used (low, medium or high viscosity) is missing, and it is known that the elastic-plastic properties of alginates are influenced by the active groups, i.e. of the viscosity of the resulting solution.
Why is the 2% concentration omitted? There are studies that show that the optimal concentration of alginate, at which good elastic properties of the capsules are obtained, is about 2%.
Lacking a satisfactory discussion regarding the choice of option C-2b? In this variant, the capsules can also turn out to be quite unstable, since the plastic properties of alginate increase with increasing of alginate concentration.
Fermentation results should be reported in standard EBC units, eg °B should be converted to °P.
References are not formatted according to journal requirements.

Author Response

Title: Formulation and characterization of calcium alginate capsules and different strains of the yeast Saccharomyces cerevisiae: impact on the sensory profile of fermented products obtained from a base malt wort.

Response to reviewers:

The authors wish to thank the reviewers and editor-in-chief for their insightful comments on the manuscript. The original comments from the reviewers are presented in bold below. Changes to the manuscript are described and highlighted with the help of Microsoft Word's tracking changes feature.

Reviewer 1

  1. What is the reason for pre-culturing the lyophilized strains? The easier way is to put the lyophilized culture directly into the alginate.

The commercial strains are hydrated according to the manufacturer's instructions for yeast activation and removal of the emulsifiers present in the sachet. Furthermore, due to the use of a native wild strain whose storage is not lyophilized but cryogenically stored, the pre-culture in all strains is made to maintain homogeneous encapsulating conditions between them, including cell forming units, which is intended to avoid factors that could interfere with the tests.

  1. The specification of the alginate used (low, medium or high viscosity) is missing, and it is known that the elastic-plastic properties of alginates are influenced by the active groups, i.e. of the viscosity of the resulting solution.

The alginate used in this study has a low M/G ratio of approximately 0.4 and a high viscosity. Section 2.1.2 now includes this information.

  1. Why is the 2% concentration omitted? There are studies that show that the optimal concentration of alginate, at which good elastic properties of the capsules are obtained, is about 2%.

According to Yadav et al. (Ref. 24), the 1.5 percent alginate, 3.5 mm diameter encapsulates performed best in the fermentative process when considering the intended application and use for the capsules. Other references, however, indicated that capsules made with a concentration of alginate less than 1.5 percent may not produce sufficiently spherical forms. Such studies also stated that the resulting encapsulates were spherical enough in the range of 2 to 5% alginate, so a concentration value in that range was chosen to evaluate the properties of the encapsulates while maintaining a diameter of 3.5 mm.

  1. Lacking a satisfactory discussion regarding the choice of option C-2b? In this variant, the capsules can also turn out to be quite unstable, since the plastic properties of alginate increase with increasing of alginate concentration.

After a few days in the culture medium (wort), the encapsulates made with 1.5% alginate showed some surface fractures and structural changes during the swelling test. As a result, they were disregarded for use in the subsequent stage of the study (fermentations). The selection of the 3% alginate capsules was made based on the breaking force and capsule morphology results.

  1. Fermentation results should be reported in standard EBC units, eg °B should be converted to °P.

The text is modified according to the reviewer's suggestion.

  1. References are not formatted according to journal requirements.

References were modified to meet the journal requirements.

Reviewer 2 Report

The manuscript entitled "The impact of yeast encapsulation in wort fermentation and beer flavor profile" submitted by Angie D. Bolaños-Barbosa, Cristian F. Rodriguez, Olga Lucía Acuña, Juan C. Cruz and Luis H. Reyes is a novelty and well-organized study. The issue addressed in the paper is actual, the study is well documented and conducted, rich in experimental results and their interpretation. The authors have made a very good job in the research of improving the beer’s sensory profile by fermenting, using different strains of S. cerevisiae yeast encapsulated in sodium alginate. The work is extensive and describes in detail the formulation and production of yeast encapsulates. The prepared capsules were characterized by multiple techniques: FTIR spectroscopy, TGA and DSC analysis, SEM, yeast cell viability etc. The second part of the work describes the fermentative processes involving encapsulated and free yeast. In each case, samples obtained after fermentation and maturation were sensory, chromatographically, and physically analyzed. As well, the authors conducted three repeated batches to evaluate the potential reusability of the encapsulates.

In conclusion, the manuscript is well written and the results are clearly presented, the findings were sufficient compared with other researches in this area, therefore it can be recommended for publishing in the Polymers journal, after a minor revision.

I suggest the following corrections in terms of editing:

1.     Some keywords are mentioned in the title. I recommend replacing them with other words that increase the visibility of the article.

2.     References should be indicated by numerals in square brackets, e.g., [5,6] instead of [5], [6] (Line 54), or [19–23] instead of [19]-[23] (Line 119), or [14, 26–28] instead of [14],[26],[27],[28] (Line 133). See also L86, L92, L130, L136 etc.

3.     L164: Lesaffre instead of Lasaffre.

4.     Why didn't you add hops to the wort before fermentation? Not to complicate the spectral analyzes and the sensory profile of the beer samples? For the industrial application of the results of your research, the beer recipe should also contain hops. Do you think that this could influence the behavior of the capsules during the alcoholic fermentation?

5.     Please revise the references list, as they should be written according to instructions for authors.

Author Response

Title: Formulation and characterization of calcium alginate capsules and different strains of the yeast Saccharomyces cerevisiae: impact on the sensory profile of fermented products obtained from a base malt wort.

Response to reviewers:

The authors wish to thank the reviewers and editor-in-chief for their insightful comments on the manuscript. The original comments from the reviewers are presented in bold below. Changes to the manuscript are described and highlighted with the help of Microsoft Word's tracking changes feature.

Reviewer 2

The manuscript entitled "The impact of yeast encapsulation in wort fermentation and beer flavor profile" submitted by Angie D. Bolaños-Barbosa, Cristian F. Rodriguez, Olga Lucía Acuña, Juan C. Cruz and Luis H. Reyes is a novelty and well-organized study. The issue addressed in the paper is actual, the study is well documented and conducted, rich in experimental results and their interpretation. The authors have made a very good job in the research of improving the beer’s sensory profile by fermenting, using different strains of S. cerevisiae yeast encapsulated in sodium alginate. The work is extensive and describes in detail the formulation and production of yeast encapsulates. The prepared capsules were characterized by multiple techniques: FTIR spectroscopy, TGA and DSC analysis, SEM, yeast cell viability etc. The second part of the work describes the fermentative processes involving encapsulated and free yeast. In each case, samples obtained after fermentation and maturation were sensory, chromatographically, and physically analyzed. As well, the authors conducted three repeated batches to evaluate the potential reusability of the encapsulates.

In conclusion, the manuscript is well written and the results are clearly presented, the findings were sufficient compared with other researches in this area, therefore it can be recommended for publishing in the Polymers journal, after a minor revision.

I suggest the following corrections in terms of editing:

  1. Some keywords are mentioned in the title. I recommend replacing them with other words that increase the visibility of the article.

Thank you for the suggestion. We modified slightly the keywords.

  1. References should be indicated by numerals in square brackets, e.g., [5,6] instead of [5], [6] (Line 54), or [19–23] instead of [19]-[23] (Line 119), or [14, 26–28] instead of [14],[26],[27],[28] (Line 133). See also L86, L92, L130, L136 etc.

The corresponding corrections were made in the document.

  1. L164: Lesaffre instead of Lasaffre.

The corresponding corrections were made in the document.

  1. Why didn't you add hops to the wort before fermentation? Not to complicate the spectral analyzes and the sensory profile of the beer samples? For the industrial application of the results of your research, the beer recipe should also contain hops. Do you think that this could influence the behavior of the capsules during the alcoholic fermentation?

Hops were not incorporated into the formulation in this initial approach to avoid masking the effects on the sensory profile caused by the yeast and its encapsulates with their flavor and aromatic contributions. The addition of hops, on the other hand, is left as a suggestion for future research.

  1. Please revise the references list, as they should be written according to instructions for authors.

References were modified to meet the journal requirements.

Reviewer 3 Report

This work is focused on the impact of encapsulation on worth fermentation and beer aroma profile. The experimental design is effective with a more than exhaustive amount of data. The main problem is that the structure of the article is complex and does not enhance the results obtained, sometimes losing focus. Thus, the work could be interesting but it should be carefully revised to focus the novelty of the results obtained. 

Please the abstract should be revised, the author could summarize the first part (Line 12-16). Moreover, the author should underline the aim of this work and the main results obtained. 

The introduction should be summarized for make the reading more useful. In my opinion the introduction does not allow to contextualize the present work.

Please delete line 36-42 and 72-86

The material and method should be summarized. Please if is possible the authors could cite others works to describe these parts briefly (i.e. Line 163-184 2.1.1. Microorganism and culture media)

The authors could add a table of code number and origin of strains tested.

The figure 1 and figure 3 are useful but could be add in supplementary material to facilitate reading.

The discussion is well structured but some hypotheses required a support many hypotheses formulated require support with the data present in the literature to increase the robustness of the statements present in the work:

Line 586-587

Line 637-640

Line: 696-697

Line:715- 717

The data obtained by electronic nose and electronic tongue are very interesting. The way chosen to represent these data does not exalt the work. Why did the authors not perform a PCA? (fig 19, 20, 21). The work could achieve benefit from it. Moreover, the PCA analysis allow to summarize the results obtained and facilitate reading and discussion of data.

Line 607 please explain why the analysis was performed only on IU-U05, FC1, FC-2 and FC-3

Figure 15. why the axis start from 0.5? Moreover, please use “Log” and not “Log10” or LOG10

The figure 18 shows the consumption of sugars and figure 16 shows BRIX°. Maybe, the authors could and figure 16 in supplementary materials and discuss only the figure 18.

Table 15. Why do the authors not perform statistical analysis? (i.e. ANOVA) This part requires a more critical discussion of data obtained

Immobilized cells compared to free cells have different behaviors as known from the literature. This work also seems to support this thesis. In order to enhance this work, a deeper discussion of these aspects would be appropriate. Please consider:

 https://doi.org/10.1016/0141-0229(94)90150-3;

https://doi.org/10.1002/yea.3042,

https://doi.org/10.1016/j.foodchem.2019.125174,

https://doi.org/10.3389/fmicb.2021.736789,

https://doi.org/10.1016/j.foodres.2021.110772

Author Response

Title: Formulation and characterization of calcium alginate capsules and different strains of the yeast Saccharomyces cerevisiae: impact on the sensory profile of fermented products obtained from a base malt wort.

Response to reviewers:

The authors wish to thank the reviewers and editor-in-chief for their insightful comments on the manuscript. The original comments from the reviewers are presented in bold below. Changes to the manuscript are described and highlighted with the help of Microsoft Word's tracking changes feature.

Reviewer 3.

This work is focused on the impact of encapsulation on worth fermentation and beer aroma profile. The experimental design is effective with a more than exhaustive amount of data. The main problem is that the structure of the article is complex and does not enhance the results obtained, sometimes losing focus. Thus, the work could be interesting but it should be carefully revised to focus the novelty of the results obtained. 

Please the abstract should be revised, the author could summarize the first part (Line 12-16). Moreover, the author should underline the aim of this work and the main results obtained. 

The introduction should be summarized for make the reading more useful. In my opinion the introduction does not allow to contextualize the present work.

  1. Please delete line 36-42 and 72-86

Only the highlighted lines 36-42 were eliminated according to the author’s criteria.

  1. The material and method should be summarized. Please if is possible the authors could cite others works to describe these parts briefly (i.e. Line 163-184 2.1.1. Microorganism and culture media)

This section was modified following suggestions 3 and 4.

  1. The authors could add a table of code number and origin of strains tested.

The table was added to section 2.1.1.

  1. The figure 1 and figure 3 are useful but could be add in supplementary material to facilitate reading.

The indicated figures were moved to the Supplementary Materials section following the reviewer's suggestion.

  1. The discussion is well structured but some hypotheses required a support many hypotheses formulated require support with the data present in the literature to increase the robustness of the statements present in the work: Line 586-587, Line 637-640, Line: 696-697, Line:715- 717.

The information was complemented in the indicated lines.

  1. The data obtained by electronic nose and electronic tongue are very interesting. The way chosen to represent these data does not exalt the work. Why did the authors not perform a PCA? (fig 19, 20, 21). The work could achieve benefit from it. Moreover, the PCA analysis allow to summarize the results obtained and facilitate reading and discussion of data.

We appreciate the reviewer's suggestion. However, the data collected in this study is insufficient to conduct a meaningful PCA that provides additional information concerning the sensory profile and therefore, we decided not to include such an analysis in the present contribution.

  1. Line 607 please explain why the analysis was performed only on IU-U05, FC1, FC-2 and FC-3

Section 3.1.6 was dedicated to test the change in yeast viability from the initial capsule to that after three repeated batches. Due to limited resources and time, however, only one strain was chosen to perform the repeated batch fermentations. The selected strain was the US05 mainly because of the manufacturer's neutral profile, which was thought to be the best option for verifying changes in the sensory profile caused solely by the presence of yeast in the fermentations. Section 3.1.6 now includes this explanation.

  1. Figure 15. why the axis start from 0.5? Moreover, please use “Log” and not “Log10” or LOG10

The recommended changes and the graphic improvements were performed according to the reviewer's suggestion.

  1. The figure 18 shows the consumption of sugars and figure 16 shows BRIX°. Maybe, the authors could and figure 16 in supplementary materials and discuss only the figure 18.

Figure 16 was moved to the Supplementary Materials section with the recommended change of units following the reviewer's suggestion.

  1. Table 15. Why do the authors not perform statistical analysis? (i.e. ANOVA) This part requires a more critical discussion of data obtained

Statistical analysis was carried out to confirm the existence of significant differences between yeast strains, as well as between free and encapsulated yeast strains. Statistical analysis was also performed on batches made with re-used capsules. Section 3.2.3 expanded on such statistical analysis.

  1. Immobilized cells compared to free cells have different behaviors as known from the literature. This work also seems to support this thesis. In order to enhance this work, a deeper discussion of these aspects would be appropriate. Please consider:

https://doi.org/10.1016/0141-0229(94)90150-3, https://doi.org/10.1002/yea.3042, https://doi.org/10.1016/j.foodchem.2019.125174, https://doi.org/10.3389/fmicb.2021.736789, https://doi.org/10.1016/j.foodres.2021.110772

The majority of the suggested papers were considered and included in sections 3.2.1, 3.2.3, and 3.1.1.

Round 2

Reviewer 1 Report

The authors take into account all suggestions and comments.

Reviewer 3 Report

The paper has been suitable for publication.